# The Impact of Working from Home on Mental Health: A Cross-Sectional Study of Canadian Worker’s Mental Health during the Third Wave of the COVID-19 Pandemic

**DOI:** 10.3390/ijerph191811588

**Published:** 2022-09-14

**Authors:** Aidan Bodner, Leo Ruhl, Emily Barr, Arti Shridhar, Shayna Skakoon-Sparling, Kiffer George Card

**Affiliations:** 1Faculty of Health Sciences, Simon Fraser University, Burnaby, BC V5A 1S6, Canada; 2Department of Psychology, Toronto Metropolitan University (Formerly Ryerson), Toronto, ON M5B 2K3, Canada; 3Institute for Social Connection, Victoria, BC V8P 5C2, Canada

**Keywords:** COVID-19, mental health, occupational health, telecommuting, masking, physical distancing

## Abstract

The COVID-19 pandemic has seen a considerable expansion in the way work settings are structured, with a continuum emerging between working fully in-person and from home. The pandemic has also exacerbated many risk factors for poor mental health in the workplace, especially in public-facing jobs. Therefore, we sought to test the potential relationship between work setting and self-rated mental health. To do so, we modeled the association of work setting (only working from home, only in-person, hybrid) on self-rated mental health (Excellent/Very Good/Good vs. Fair/Poor) in an online survey of Canadian workers during the third wave of COVID-19. The mediating effects of vaccination, masking, and distancing were explored due to the potential effect of COVID-19-related stress on mental health among those working in-person. Among 1576 workers, most reported hybrid work (77.2%). Most also reported good self-rated mental health (80.7%). Exclusive work from home (aOR: 2.79, 95%CI: 1.90, 4.07) and exclusive in-person work (aOR: 2.79, 95%CI: 1.83, 4.26) were associated with poorer self-rated mental health than hybrid work. Vaccine status mediated only a small proportion of this relationship (7%), while masking and physical distancing were not mediators. We conclude that hybrid work arrangements were associated with positive self-rated mental health. Compliance with vaccination, masking, and distancing recommendations did not meaningfully mediate this relationship.

## 1. Introduction

The COVID-19 pandemic has exacerbated many risk factors for poor mental health in the workplace. As this pandemic has intensified, with rising cases and deaths globally, so too have feelings of worry and fear in response to ongoing COVID-19 community transmission [1,2]. Studies from across the world have demonstrated that many workers are afraid of contracting and transmitting COVID-19 while at work [3,4,5,6]. Fear is an adaptive defense mechanism for humans when confronted with a risk or danger, however chronic fear can lead to adverse mental health outcomes and behaviours. In the COVID-19 pandemic, fear of COVID-19 has been associated with depression, anxiety, and even impaired job performance [5]. A Canadian study from May 2020 reported that mental health has worsened since the onset of the COVID-19 pandemic, due in large part to economic uncertainty and fear of illness [7]. Notably, these negative mental health effects have largely been observed in work settings that are predominantly public-facing and more exposed to viral transmission [3,4,5,8,9,10,11,12].

Alongside healthcare workers, many low-wage service workers have been deemed essential workers in Canada, and like other front-facing workers at the start of the pandemic, these workers have not always had access to safe working environments [3,13]. At several points in the pandemic, many workers had to attend in-person positions without widespread availability of COVID-19 vaccines or public health mandates, effectively exposing them to anxiety-provoking environments. The pandemic has also heightened burdens that impact mental health among essential workers, including: adopting caretaking roles of vulnerable family members; choosing between working through illness or taking time off and facing financial losses when sick; lower job security; reduced income; greater risk of contracting COVID-19; and slashed work hours [10,14,15,16,17]. These burdens intersect with other socio-demographic factors. For example, ethnic minorities and recent immigrants in Canada are more likely to work in low-wage, public-facing positions, which highlights health equity concerns given the increased risk for COVID-19 transmission and accompanying mental health disorders in this population [18,19].

While mental health risks are well-known among public-facing workers, it is less clear what the mental health impacts are on workers who have been able to transition to working from home. Workers at home may experience a more complex impact of their work settings on their mental health, despite having a generally lower risk situation [20,21,22]. Although much of the research studying teleworks impacts on workers mental health during the pandemic is ongoing, several studies have already shed light on this relationship. For example, some research has shown that workers who were more afraid of COVID-19 were more productive when working from home [23]. When faced with going back to in-person work, many workers anticipate negative impacts specifically due to concerns about COVID-19 safety [24]. Conversely, telework during the pandemic has also been associated with increases in social isolation and work stress [23,25], family conflict [22,23], distractions [22,23], as well as food and alcohol consumption [22,26]—which can all negatively impact the mental health of workers [22]. A recent study from Portugal has shown that employees working from home felt like they needed to appear online and in touch with their colleagues more often, correlating depression, anxiety and stress [25].

The literature exploring differences in mental health outcomes between workers in public-facing occupations and those working from home in Canada has been sparse [13,27]. One study conducted in the first half of 2020 measured anxiety and depression symptoms through Generalized Anxiety Disorder 2-item (GAD-2) and Patient Health Questionaire-2 (PHQ-2) screeners. These objective measures of mental health contribute only to a narrow understanding of mental health in relation to overall wellbeing. Similarly, most of the current research has examined telework during the first waves of COVID-19. Although useful, this work may not fully capture the impact that novel interventions such as vaccines and mask mandates have on the mental health of workers. Unlike during the first waves of the pandemic, Canadians now have access to free vaccines and masks; and other risk mitigation approaches (e.g., physical distancing, ventilation) are better understood by the public. These measures may, therefore, mitigate the fear of COVID-19 and its associated stress for people working in public, front-facing jobs [3]. Conversely, we have also experienced a slow relaxation of public health orders which enforced COVID-19 protection behaviours, such as social distancing, vaccine, and mask mandates, which may increase feelings of fear or anxiety about returning to work. Thus, there is a need to explore this area further.

Furthermore, the first doses of the vaccine rollout for the general population in Canada were underway during the third wave of the pandemic in 2021, bringing about another layer of nuance to consider when assessing mental health of [28]. This development added complexity in both negative and positive directions via the potential for increased apprehension and vaccine hesitancy, as well as the potential for reduced mental distress as a result of the sense of protection offered by the vaccine [29,30]. Reduced mental distress due to the availability of COVID-19 vaccines may have also been more likely due to the mentally taxing events of the first and second waves which saw an overwhelmed healthcare system, deaths in long-term care facilities, and socially isolating lockdown measures [31,32,33].

Presently, at the end of the sixth wave of the COVID-19 pandemic has seen jurisdictions move further away from public health orders, following roll-outs of third doses for the majority of working age adults in response to the Omicron variant [34,35]. It remains unclear how the ongoing need for vaccine uptake and the turbulent nature of the pandemic will impact mental health. Moreover, as many companies and organizations transitioned large numbers of staff to working from home or a hybrid of working from home and in-person work during earlier waves of the pandemic, this work will be relevant for both employers and policy makers respectively to assess the costs and benefits of different arrangements as workplaces largely return to in-person work. Determining the extent of any differences in mental health related to work-from-home status has clear health equity implications for employers and policy makers to ensure best practices throughout the ongoing COVID-19 pandemic, as well as for future public health crises. As COVID-19 risks continue to the present day—particularly with risks such as long-COVID and unmitigated Omicron infection—it has become important to understand mental health differences according to where participants are working.

This study used survey data collected during the third wave of the COVID-19 pandemic in Canada [36] to examine whether there were any differences in self-rated mental health based on work setting and if so, what contributes to these differences? The dataset provided a unique opportunity to explore the nuances of self-rated mental health, and thus, bivariable and multivariable logistic regression models were used to test the hypothesis that mental health status is poorer among individuals who are not working from home. Additionally, physical distancing and mask wearing, which have been common practice since the onset of the pandemic, will be tested as mediators due to their potential for combating pandemic-related stressors related to concerns about COVID-19 transmission [37]. A mediation analysis tested whether COVID-19 vaccination, physical distancing, and mask adherence—due to their effectiveness as COVID-19 mitigation measures—had significant and protective effects on self-rated mental health. In conducting these analyses, we hypothesized that people working from home or engaging in hybrid work arrangements had better self-rated mental health than those working exclusively in-person. We further hypothesized that the exposure to COVID-19, as reflected in lack of compliance with public safety COVID-19 prevention guidelines, would partially mediate the association between working from home and worse self-rated mental health.

## 2. Materials and Methods

### 2.1. Study Data

The study utilized the Canadian Social Connection Survey (CSCS) dataset, which collected data from 21 April to 1 June 2021. The survey was circulated on the internet using paid advertising on Facebook, Twitter, Instagram, and Google. Participants were eligible if they were Canadian residents and 16 years of age or older. Ethics approval was granted by the University of Victoria Research Ethics Board (Ethics Protocol Number 21-0115) [36]. All participants provided informed consent and were able to complete the questionnaire in English or French. Given the need to determine mental health effects in various work settings, the dataset allows for a comprehensive exploration. Inclusion for the current study was conditional on whether a respondent indicated that they were working during the COVID-19 pandemic.

A total of 2286 eligible participants completed the survey. Of these, 1917 were working during the COVID-19 pandemic. We excluded participants with missing observations on the primary outcome (i.e., self-rated mental health) and primary exposure variable (i.e., amount of work from home during COVID-19); thus, the analytic sample size for this analysis was 1576.

### 2.2. Study Measures

#### 2.2.1. Outcome Variable

Respondents’ self-rated mental health was the primary outcome variable for the study. This variable has previously shown a positive correlation to other mental health morbidity measures [38], but should not be conflated with other more specific diagnostic categories such as depression or anxiety [39,40]. Indeed, as a more global and subjective measure, many authors consider self-rated mental health as a more holistic measure of mental health outcomes which allows for a broad range of mental health issues to be captured [38,41], including mental health problems that are developing but which are not captured by more clinical mental health indicators [40]. Participants evaluated their current mental health on a Likert scale (At the present time, would you say your MENTAL HEALTH is: “Poor”, “Fair”, “Good”, “Very good”, or “Excellent”) (see Appendix A). The variable was dichotomized to “Negative Self-Rated Mental Health” (“Poor” and “Fair”) and “Positive Self-Rated Mental Health” (“Good”, “Very good”, and “Excellent”). This was deemed to be an acceptable (if not conservative) approach to capture a general sense of mental health status based on precedent from previous studies using self-rated mental health [38]—allowing us to explicitly identify factors associated with sub-optimal (i.e., fair or poor) mental health.

#### 2.2.2. Primary Explanatory Variable

Work setting (listed as work_from_home in the dataset) was the primary explanatory variable for the study. The variable measured how often participants worked from home (“Not Working During COVID”, “Not at all”, “Very little of the time”, “Some of the time”, “Most of the time”, and “All of the time”). The levels “Very little of the time”, “Some of the time”, and “Most of the time” were collapsed into a single level—“Hybrid”. “Not at all” was recoded as “Do Not Work from Home” and “All of the time” was recoded as “Work from Home Only”. These levels allowed for a continuum of working from home to be represented. Participants who reported not working during COVID-19 were removed from analyses as our goal was to explore the effects among Canadian workers who were currently employed.

#### 2.2.3. Confounding Variables

Other explanatory variables related to employment, adherence to COVID-19 mitigation measures, income, and identity were controlled for in multivariable analysis. This allowed us to isolate the effects of demographic and socio-economic factors which may otherwise play an important role in self-rated mental health while also being correlated with work setting. The included variables were household income (originally collected in increments of CAD 10,000, but binned into four groups capturing low, lower-middle, middle, and upper income groups: Less than CAD 30,000, CAD 30,000 to CAD 59,999, CAD 60,000 to CAD 89,999, CAD 90,000 or more), age (18 to 29 years-old, 30 to 39 years-old, 40 to 49 years-old, 50 to 59 years old, 60 years and older), gender (Male, Non-binary, Woman), ethnicity (White; African, Caribbean, or Black; Asian; Indigenous; Middle Eastern; Other), educational attainment (High School Diploma or Lower, Bachelor’s Degree or Higher, Some College), hours worked per week (participant-reported numeric value), national occupation class (Art, culture, recreation and sport; Business; Education, law and social, community, and government services; Health; Management; Manufacturing and utilities; Natural and applied sciences; Natural resources and agriculture; Sales and service; Trades, transport and equipment operators).

In addition to these conventional confounding variables, several additional variables were selected based on their potential to mediate the relationship between self-reported mental health and work setting. COVID-19 vaccine status and adherence to mask and/or physical distancing recommendations were identified as particularly important factors with mediation potential. These concepts were measured by asking to what extent participants wore masks in public (“Not at all”, “Somewhat”, “Very Closely”), to what extent participants practice physical distancing in public (“Not at all”, “Somewhat”, “Very Closely”), and whether participants were vaccinated (“No”, “Yes, one dose”, “Yes, two doses”).

### 2.3. Statistical Analysis

All statistical analyses were performed using R Statistical Software version 4.1.1 (R Foundation for Statistical Computing, Vienna, Austria) [42]; DescTools and regclass packages were used to assist in model assessment and fitting [43,44]; the mice package was used for multiple imputations of missing observations [45]; and the mediation package was used for mediation analysis [46]. Missing observations on the remaining variables were imputed using multiple imputation in the mice package [45].

An initial multivariable binary logistic regression model (Appendix A), with the outcome variable of self-rated mental health and primary explanatory variable of work setting, was constructed with 30 confounding variables. The final multivariable model was developed by running a backwards selection process favouring the model with lowest Akaike Information Criterion [47]. This process was balanced by supplementing the model with variables critical to understanding the relationship between work-setting and self-rated mental health that the backwards selection process had excluded. McFadden’s Pseudo R^2^ and variance inflation factor were assessed for reasonability of model fit and collinearity, with variables exhibiting collinearity removed to arrive at a final multivariable model. Bivariable logistic regression models were constructed from the newly developed study sample between all explanatory variables and the outcome variable.

Mediation analysis was followed firstly via Baron and Kenney’s (1986) steps for determining mediation via logistic regression models and secondly by utilizing the mediate package in R with bootstrapping enabled [48,49]. The mediate package explicitly allows for handling of binary and logistic measures outside of a linear framework, while Baron and Kenney’s (1986) steps provide a process for reviewing bivariable and multivariable models, which has helped us to evaluate the associations between our primary exposure and outcome, primary exposure and mediator, mediator and outcome, and primary exposure while controlling for the mediator and outcome. The mediate function was then used for more rigorous tests of indirect (mediation) effects on the outcome variable [49].

## 3. Results

### 3.1. Sample Overview

2286 respondents were initially included. However, 370 indicated they were not currently employed and of the remaining 1916 employed respondents, 340 were missing data on our primary measures. This resulted in 1576 participants eligible for analysis. Descriptive statistics, stratified by self-rated mental health, are presented in Table 1. The study sample predominantly reported positive self-rated mental health (80.7%) with the majority of participants in both outcome groups responding that they work both from home and in person (hybrid); however, a greater proportion (46%) of those not working from home reported negative self-rated mental health compared to those in other work setting configurations (Figure 1). In terms of demographics, 41.8% were 18 to 29 years-old; 49.9% identified as a man; 65.5% were White; 36.0% earned between CAD 30,000 and CAD 59,000 in 2020; and 51.0% had a Bachelor’s degree or higher. The average number of reported hours worked per week was 23.87; 19.9% worked in sales and service; 53.7% indicated they very closely practice physically distancing 2 metres from others; 72.8% reported very closely adhering to wearing masks in public; and 56.8% had received one dose of a COVID-19 vaccine.

### 3.2. Regression Analysis

Bivariable associations were investigated between all explanatory variables and self-rated mental health (Table 2). Associations between self-rated mental health and work setting were significant among people not working from home as well as those exclusively working from home. These groups had respectively 5.70 (95% Confidence Interval [95% CI]: 3.98, 8.15) and 3.97 (95% CI: 2.85, 5.52) greater odds of negative self-rated mental health as compared to people working in hybrid arrangements. Other significant bivariable associations with negative self-rated mental health were age (all ages over 40 years-old versus those 18 to 29 years-old) and being non-binary or a woman (vs. a man). Positive self-rated mental health was significantly associated with African, Caribbean, or Black ethnicity (vs. White) and Indigenous ethnicity (vs. White); having some college education or a Bachelor’s degree or higher (vs. high school diploma or lower); employment in business, health, management, natural and applied sciences, or trades, transport and equipment operations (vs. sales and services); and having one or two doses of a COVID-19 vaccine (vs. not having received a COVID-19 vaccine).

In the multivariable model, after controlling for potential confounders, negative self-rated mental health retained the association with not working from home (Adjusted Odds Ratio [aOR]: 2.79, 95% CI: 1.83, 4.26) and working from home exclusively (aOR: 2.79, 95% CI: 1.90, 4.07) versus hybrid work. Furthermore, negative self-rated mental health was significantly associated with increasing hours worked per week, being 40 years or older (vs. 18 to 29 years-old), identifying as non-binary (vs. man), Middle Eastern or Other ethnicity (vs. White), Conversely, positive self-rated mental health was associated with employment in business, health, management, natural and applied sciences, or trades, transport and equipment operations (vs. sales and services); and having two doses of a COVID-19 vaccine (vs. not having received any).

### 3.3. Mediation Analysis

Table 3 illustrates the results of the mediation analyses for each of the three COVID-19 prevention factors. Vaccination status was found to be a statistically significant mediator (*p* = 0.02), mediating approximately 7% of the relationship between work setting and self-rated mental health; mask wearing (*p* = 0.76) and physical distancing (*p* = 0.20) were not found to significantly mediate the relationship. In the mediation analyses for vaccination status, the first part of the pathway between work setting and self-rated mental health, when adjusting for having received a COVID-19 vaccine, shows not working from home is significantly associated with negative self-rated mental health (aOR: 3.91, 95% CI: 2.74, 5.56). The next part of the pathway between work setting and having received a COVID-19 vaccine indicates people not working from home had lower odds of having at least one dose of a COVID-19 vaccine (OR: 0.52, 95% CI: 0.39, 0.70). The last part of the pathway shows a significant association between having received a COVID-19 vaccine and positive self-rated mental health (OR: 0.30, 95% CI: 0.21, 0.43).

## 4. Discussion

### Primary Findings

This study represents a preliminary assessment of the relationship between work setting and self-rated mental health, controlling for relevant demographic factors, and providing several preliminary insights into the ways in which COVID-19 stressors and protections shape these relationships. In doing so, our findings show that mental health is adversely impacted for those either working exclusively from home or in person. This is in agreement with existing literature showing poor mental health among workers in public-facing workspaces across numerous international contexts [8,9,10,11,12,13,14]. Similarly, although findings of studies examining mental health effects of working from home prior to the COVID-19 pandemic have been inconsistent [21], studies exploring this increasingly normalized work setting during the pandemic have generally found working from home associated with poorer mental health outcomes [26]. This is often attributed to difficulties in establishing a work-life balance and due to feelings of isolation [22,23,50,51]. However, the current findings are unique in that only a handful of studies investigating the link between workplace and mental health during COVID-19 to-date have directly examined varying degrees of working from home [8,9,13,27] and none to our knowledge have investigated these associations during the later phases of the COVID-19 pandemic, when vaccines were made widely available. Furthermore, the majority of studies have explored the mental health of healthcare workers [2,11,12,52] or those in public-facing positions [10]. As such, the present study makes a valuable contribution in terms of the timing within the COVID-19 pandemic, its focus on a broad range of labour sectors, and its use of holistic self-rated mental health measures.

As such, these findings help to further research into the mental health outcomes of the Canadian workforce during the later phases of ongoing COVID-19 pandemic and beyond. One Canadian study exploring the relationship between working from home and self-rated mental health (although not of primary interest) during the first wave of the pandemic found that workers who transitioned to working from home did not differ or have affected mental health when compared to those who remained working in-person. Conversely, another Canadian study from the first wave of the pandemic found lower prevalence of depression and anxiety among respondents working from home or those working in person whose employers met all of their infection control needs [27]. These findings differ from what this study has found during the third wave, namely: both not working from home and working exclusively from home are significantly associated with negative self-rated mental health. Turning to international evidence (again from the first wave), both Gómez-Salgado et al. (2020) and Mazza et al. (2020) found poorer mental health was associated with not working from home, when compared to working from home, and not working at all, respectively. The range of evidence adds credence to our findings indicating negative mental health outcomes at either end of the work from home continuum—where workers are exclusively working from one location.

The mediation analysis found that, of the three variables tested, COVID-19 vaccination status was the only significant mediator of the effect of work setting on self-rated mental health. However, this variable mediated only approximately 7% of the effect of work setting on self-rated mental health. Both the lack of significance and the low impact of the mediation among the variables tested suggests that the prominent source of psychological stress may not arise from fear of COVID-19 infection. Although it is likely that these prevention measures may do less to mediate mental health among workers who are not continually facing risk of viral exposure, it is less clear why this would also be the case for public-facing workers. One possibility could be that, by the later phases of the COVID-19 pandemic, workplaces already tended to have high levels of COVID-19 control measures in place [53], likely reducing the contribution of the environment to stress related to concerns about viral exposure. Secondly, views on the severity of COVID-19 symptoms or susceptibility to it may have an impact on the extent that the COVID-19 prevention measures mediate mental health [54]. Lastly, uncertainty related to the unpredictable trajectory of the pandemic, such as economic concerns may present as greater stressors when compared to fears of COVID-19 infection [55].

This study also highlighted poor negative mental health among several groups. Though we did not specifically explore groups that are more likely to work from home, concerns have been raised about the well-being of ethnic minority groups who disproportionately work in public-facing occupations [56]. These sectors have experienced numerous disruptions in their capacity to operate throughout the COVID-19 pandemic [19]. This has had severe effects on members of ethnic minorities. For instance, in mid-2020, 44% and 40% of people of Arabic and West Asian ethnicity respectively, reported that the COVID-19 pandemic had moderate to strong impacts on their financial stability [57].

The identity groups associated with negative self-rated mental health—non-binary individuals and people over 40 years—are less clear in terms of contextualizing within work setting. For non-binary individuals, it is unclear whether they are more likely to work from home; however, it does appear that the pre-pandemic stressors have been compounded by COVID-19 for members of sexual and gender minorities [58]. As for middle-and-older age workers, the association with negative self-rated mental health corresponds to a general trend that mental health has worsened for all age groups in Canada since the onset of the pandemic [59]; however, it is unclear what this finding may mean in the context of other studies, indicating better mental health among older adults during the pandemic [60,61].

Despite COVID-19 prevention measures not emerging as a primary influencer of self-rated mental health, Canadian provinces such as British Columbia have routinely made it a priority to vaccinate frontline workers, a category of worker who cannot typically work from home [62]. Moreover, in examining other sources of economic-related stress, initial pandemic responses did see the Canadian federal government initiating supports for unemployed workers such as the Canada Emergency Response Benefit (CERB) in conjunction with provincial eviction bans, and to a lesser extent, rent freezes [63]. Though CERB provided support for workers financially impacted by the pandemic, workers who continued to be employed did not enjoy these benefits, despite facing the possibility of reduced work hours. Moreover, rent freezes that were widely enacted by provincial governments were largely discontinued after December 2020 [63]. Thus, despite a relatively rapid implementation of social protections in response to the arrival of COVID-19 in Canada [64], the lack of continuity of these measures coupled with pandemic uncertainty may feed into stressors affecting Canadian workers.

## 5. Limitations

This exploratory study has limitations but provides rationale for more rigorous investigations of the potential benefits of hybrid work. Limitations include our use of secondary data that likely does not fully capture the nuanced associations between work setting and self-rated mental health. These relationships are further simplified by our analytic choices to collapse work setting to three levels and self-rated mental health to two levels. Future studies should explore more comprehensive measures of mental health, including using specific measures of anxiety and depression. Such analyses might be feasible in large surveys, such as ours, through the use of short scales developed for large surveys, such as the PHQ-2 and GAD-2. It is possible that these more specific measures would allow for greater granularity in understanding how working conditions during an ongoing public health crisis is related to mental health and well-being—particularly in terms of the mediating effects of COVID-19 prevention on anxiety and stress (vs. depression). Qualitative research could also be used to better understand specific pathways of poor mental health for those working exclusively from home or in-person. Given limitations in measurement, the results of the current study must be interpreted with caution when considering specific psychological disorders. As well, the dataset over-represented (77.2%) individuals who work in hybrid arrangements, compared to the other two groups (exclusively working from home and exclusively working in-person). Caution should therefore be taken in interpretation, as this drastically departs from the range of Canadian workers working the majority of their hours from home—40.5% in April 2020 to 26.5% in June 2021 [65]. Lastly, as the CSCS did not include questions assessing individuals’ worry about COVID-19 exposure at work, nor how well their workplace implemented protection protocols, we were not able to account for the nuance of psychological distress related to COVID-19 infection. The measures we use to assess compliance are global and not work specific. As such, our mediation models should be interpreted as preliminary. Likewise, some measures need refined assessment in future studies. For example, to measure income, participants’ household incomes were collected in increments of $10,000 CAD. Bins of $30,000 CAD were selected with consideration of classifying individuals according to approximate thresholds for low- (e.g., Approx. $30,000 per households) and median income (approx. $90,000 per household) in Canada. As household size and cost-of living values varied, a more nuanced measure of income would have been preferred by was not available in this secondary data analysis. Personal income, adjusted for cost of living, could provide a more nuanced insight into working condition and types of work engaged in, as these parameters are undoubtedly important for understanding worker health.

## 6. Future Research Directions

Recognizing these limitations, as well as several opportunities to establish new lines of inquiry, we recommend that future research on the COVID-19 pandemic and future communicable disease epidemics should aim to sample a more representative group of people working from home; determine interactions between ethnic, sexual and gender minorities, and older populations; and incorporate measures of self-assessed psychological distress around workplace safety. Furthermore, as noted above, the present study did not account for important and salient factors such as living conditions, household composition, sources of material, social, and emotional support, non-work-related labor, and other undoubtedly important factors. Future research will explore these factors in relation to working arrangements. Such analyses are critical for understanding the gendered dynamics of work from home. We hypothesize that this would be a critical moderator for exploration in future research. As well, family composition and income are critical moderators for understanding how people can best be supported in distance work environments. Therefore, future research should conduct more narrow analyses or improve measurements of these key factors so that a more nuanced profile of working conditions (e.g., income, class, status, hierarchy) can be assessed in relation to our research questions. Finally, it is critical for longitudinal within person studies to continue examining the effect of work from home on individual health and wellbeing.

## 7. Conclusions

Given the few studies that are available assessing the effect of work setting on mental health, this study provides important data demonstrating potential hazards to mental health associated with exclusively in-person or home-based work. Hybrid models of work may therefore provide promising opportunities to improve the mental health of workers. Of course, replication will further advance our understanding of telecommuting and in-person work, particularly in the context of an ongoing public health crisis that has disproportionately impacted low-wage and marginalized people.

## Figures and Tables

**Figure 1 ijerph-19-11588-f001:**
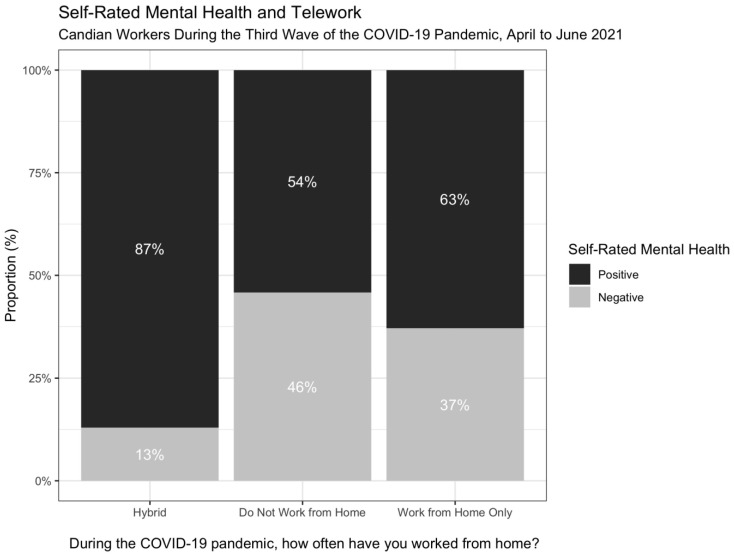
Work Setting and Self-Rated Mental Health.

**Table 1 ijerph-19-11588-t001:** Sample Characteristics Stratified by Self-Rated Mental Health.

	Overall	Positive Self-Rated Mental Health	Negative Self-Rated Mental Health	*p*-Value
**N (%)**	1576 (100)	1272 (80.7)	304 (19.3)	
**Age (Years)**				
18 to 29 years-old	658 (41.8)	572 (45.0)	86 (28.3)	<0.001
30 to 39 years-old	543 (34.5)	460 (36.2)	83 (27.3)	
40 to 49 years-old	169 (10.7)	115 (9.0)	54 (17.8)	
50 to 59 years-old	119 (7.6)	71 (5.6)	48 (15.8)	
60 years and older	87 (5.5)	54 (4.2)	33 (10.9)	
**Gender**				<0.001
Man	787 (49.9)	666 (52.4)	121 (39.8)	
Non-binary	41 (2.6)	26 (2.0)	15 (4.9)	
Woman	748 (47.5)	580 (45.6)	168 (55.3)	
**Ethnicity**				0.0024
White	1033 (65.5)	824 (64.8)	209 (68.8)	
African, Caribbean, or Black	158 (10.0)	141 (11.1)	17 (5.6)	
Asian	132 (8.4)	105 (8.3)	27 (8.9)	
Indigenous	103 (6.5)	92 (7.2)	11 (3.6)	
Middle Eastern	45 (2.9)	34 (2.7)	11 (3.6)	
Other	105 (6.7)	76 (6.0)	29 (9.5)	
**Household Income**				0.8181
Less than CAD 30,000	474 (30.1)	382 (30.0)	92 (30.3)	
CAD 30,000 to CAD 59,999	567 (36.0)	461 (36.2)	106 (34.9)	
CAD 60,000 to CAD 89,999	376 (23.9)	305 (24.0)	71 (23.4)	
CAD 90,000 or more	159 (10.1)	124 (9.7)	35 (11.5)	
**Educational Attainment**				0.013
High School Diploma or Lower	187 (11.9)	136 (10.7)	51 (16.8)	
Bachelor’s Degree or Higher	804 (51.0)	657 (51.7)	147 (48.4)	
Some College	585 (37.1)	479 (37.7)	106 (34.9)	
**Hours Worked per Week**	23.87 (16.8)	22.48 (16.54)	29.70 (16.74)	<0.0001
**National Occupation Class**				<0.0001
Sales and Service	313 (19.9)	234 (18.4)	79 (26.0)	
Art, Culture, Recreation and sport	102 (6.5)	82 (6.4)	20 (6.6)	
Business	228 (14.5)	195 (15.3)	33 (10.9)	
Education, Law and Social, Community, and Government Services	272 (17.3)	195 (15.3)	77 (25.3)	
Health	180 (11.4)	152 (11.9)	28 (9.2)	
Management	193 (12.2)	168 (13.2)	25 (8.2)	
Manufacturing and utilities	47 (3.0)	37 (2.9)	10 (3.3)	
Natural and applied sciences	103 (6.5)	93 (7.3)	10 (3.3)	
Natural resources and agriculture	48 (3.0)	37 (2.9)	11 (3.6)	
Trades, transport and equipment operators	90 (5.7)	79 (6.2)	11 (3.6)	
**Work Setting**				<0.0001
Hybrid	1216 (77.2)	1059 (83.3)	157 (51.6)	
Do Not Work from Home	155 (9.8)	84 (6.6)	71 (23.4)	
Work from Home Only	205 (13.0)	129 (10.1)	76 (25.0)	
**COVID-19 Guideline Adherence: Distancing from Others By 2 Meters or More**				0.6947
Not at all	89 (5.6)	74 (5.8)	15 (4.9)	
Somewhat	641 (40.7)	521 (41.0)	120 (39.5)	
Very Closely	846 (53.7)	677 (53.2)	169 (55.6)	
**COVID-19 Guideline Adherence: Wearing Masks**				0.0067
Not at all	60 (3.8)	50 (3.9)	10 (3.3)	
Somewhat	369 (23.4)	318 (25.0)	51 (16.8)	
Very Closely	1147 (72.8)	904 (71.1)	243 (79.9)	
**Vaccination Status**				<0.0001
No	286 (18.1)	204 (16.0)	82 (27.0)	
Yes, one dose	895 (56.8)	725 (57.0)	170 (55.9)	
Yes, two doses	395 (25.1)	343 (27.0)	52 (17.1)	

**Table 2 ijerph-19-11588-t002:** Bivariable and Multivariable Logistic Regression Models.

		Bivariable			Multivariable	
		95% CI			95% CI	
	OR	Lower	Upper	aOR	Lower	Upper
**Work Setting (Ref = Hybrid)**						
Do Not Work from Home	**5.70**	**3.98**	**8.15**	**2.79**	**1.83**	**4.26**
Work from Home Only	**3.97**	**2.85**	**5.52**	**2.79**	**1.90**	**4.07**
**Hours Worked per Week**	1.03	1.02	1.03	**1.02**	**1.01**	**1.03**
**Income (Ref = Less than CAD 30,000)**						
CAD 30,000 to CAD 59,999	0.95	0.70	1.30	0.77	0.53	1.10
CAD 60,000 to CAD 89,999	0.97	0.68	1.36	0.78	0.53	1.16
CAD 90,000 or more	1.17	0.75	1.80	0.93	0.56	1.52
**Age (Ref = 18 to 29 years-old)**						
30 to 39 years-old	1.20	0.87	1.66	1.19	0.83	1.71
40 to 49 years-old	**3.12**	**2.10**	**4.63**	**2.31**	**1.48**	**3.59**
50 to 59 years-old	**4.50**	**2.92**	**6.91**	**2.93**	**1.76**	**4.85**
60 years and older	**4.06**	**2.48**	**6.61**	**2.47**	**1.39**	**4.36**
**Gender (Ref = Man)**						
Non-binary	**3.18**	**1.60**	**6.10**	**2.61**	**1.19**	**5.54**
Woman	**1.59**	**1.23**	**2.07**	1.15	0.86	1.56
**Ethnicity (Ref = White)**						
African, Caribbean, or Black	**0.48**	**0.27**	**0.78**	0.79	0.43	1.38
Asian	1.01	0.64	1.57	1.11	0.66	1.82
Indigenous	**0.47**	**0.23**	**0.86**	0.84	0.40	1.61
Middle Eastern	1.28	0.61	2.48	**2.69**	**1.18**	**5.76**
Other	1.50	0.94	2.34	**1.96**	**1.16**	**3.27**
**Educational Attainment (Ref = High School Diploma or Lower)**						
Bachelor’s Degree or Higher	**0.60**	**0.41**	**0.87**	0.70	0.45	1.10
Some College	**0.59**	**0.40**	**0.87**	0.76	0.49	1.20
**National Occupation Class (Ref = Sales and service)**						
Art, Culture, Recreation and sport	0.72	0.41	1.23	0.82	0.44	1.49
Business	**0.50**	**0.32**	**0.78**	**0.57**	**0.34**	**0.94**
Education, Law and Social, Community, and Government Services	1.17	0.81	1.69	0.86	0.55	1.34
Health	**0.55**	**0.33**	**0.87**	**0.47**	**0.28**	**0.80**
Management	**0.44**	**0.27**	**0.71**	**0.47**	**0.27**	**0.80**
Manufacturing and utilities	0.80	0.36	1.63	0.78	0.33	1.71
Natural and applied sciences	**0.32**	**0.15**	**0.61**	**0.42**	**0.19**	**0.85**
Natural resources and agriculture	0.88	0.41	1.76	0.74	0.31	1.66
Trades, transport and equipment operators	**0.41**	**0.20**	**0.78**	**0.34**	**0.15**	**0.70**
**Wears Mask in Public (Ref = Not at All)**						
Somewhat	0.80	0.40	1.77	1.02	0.45	2.46
Very Closely	1.34	0.70	2.85	1.58	0.71	3.79
**Practices Physical Distancing in Public (Ref = Not at All)**						
Somewhat	1.14	0.65	2.12	1.32	0.66	2.78
Very Closely	1.23	0.71	2.28	1.02	0.51	2.20
**Is Vaccinated (Ref = No)**						
Yes, one dose	**0.58**	**0.43**	**0.79**	0.71	0.50	1.02
Yes, two doses	**0.38**	**0.25**	**0.55**	**0.56**	**0.36**	**0.87**

Numeric bolding: Indicates statistical significance.

**Table 3 ijerph-19-11588-t003:** Relationship between Work Setting (Ref = At least some of the time (Hybrid/Work from home only)), Mediators (Vaccination Status (Ref = No), Adherence to Mask Wearing Recommendations (Ref = Not at all), and Adherence to Physical Distancing Recommendations (Ref = Not at all)), and Self-Rated Mental Health (Ref = Positive).

	Vaccination Status	Mask Wearing	Physical Distancing
WS → Vaccination ^1^	**0.30 (0.21, 0.43)**		
Vaccination → SRMH ^1^	**0.52 (0.39, 0.70)**		
WS → SRMH ^2^	**3.91 (2.74, 5.56)**		
Proportion Mediated (Average)	**0.07 ***		
WS → Masks ^1^		0.82 (0.40, 2.00)	
Masks → SRMH ^1^		1.20 (0.63, 2.54)	
WS → SRMH ^2^		4.32 (3.05, 6.10)	
Proportion Mediated (Average)		−0.002	
WS → Distancing ^1^			0.47 (0.27, 0.86)
Distancing → SRMH ^1^			1.19 (0.69, 2.18)
WS → SRMH ^2^			4.40 (3.10, 6.22)
Proportion Mediated (Average)			−0.01

^1^ OR = Odds Ratio (95% Confidence Interval); ^2^ aOR = Adjusted Odds Ratio (95% Confidence Interval); * *p* ≤ 0.05; Numeric bolding: Indicates statistical significance; WS = Work setting; SRMH = Self-rated mental health.

## Data Availability

Data used in the study analysis is stored and available on the OSF Repository (https://osf.io/87vgs/, accessed on 3 August 2022).

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
