# Peer review of "The Impact of Working from Home on Mental Health: A Cross-Sectional Study of Canadian Worker’s Mental Health during the Third Wave of the COVID-19 Pandemic"

_ijerph, 2022, doi:10.3390/ijerph191811588_

Round 1
Reviewer 1 Report
This well-written paper deals with a relevant topic: the impact of working from home on mental health during the third wave of the covid-19 pandemic. This article may be appropriate for publication with minor revisions of the following points:
• (line 139) be more precise about the primary outcome: the primary outcome is the Self-rated general sense of mental health status (it could be misleading to speak of "mental health" as the correlations with specific outcome measures (anxiety, depression , etc,) are not that significant);
• (line 141-142) the cited article (Levinson et al, 2014) does not show such overwhelming significance, so very cautious in this statement (remove it to avoid overestimating this association with the DSM categories);
• (line 146) report the question asked to the participants to evaluate the general sense of the state of mental health
• (line 241-242) are you sure that there is a significant association with such a low percentage of "non-binary" participants (2.6%) "Other significant bivariable associations with negative self-assessed mental health were [] being not binary or a woman (against a man). "): check the analyzes again
• (242-243) are you sure that there is a significant association with such a low percentage of "African, Caribbean, or Black ethnicity" participants (2%) (the text states "self-rated mental health was significantly associated with African, Caribbean or black (vs. white) ethnicity "): check the analyzes again
Author Response
(line 139) be more precise about the primary outcome: the primary outcome is the Self-rated general sense of mental health status (it could be misleading to speak of "mental health" as the correlations with specific outcome measures (anxiety, depression, etc,) are not that significant). Further, (line 141-142) the cited article (Levinson et al, 2014) does not show such overwhelming significance, so very cautious in this statement (remove it to avoid overestimating this association with the DSM categories)
Response: Thank you for noting this issue. We have added a new citation (from Ahmad et al. (2014) that provides a scoping review of this measure. As well, we have referred to our outcome as self-rated mental health where appropriate and provided the following discussion to clarify the distinctions between self-rated mental health and DSM diagnoses:
“Respondent’s self-rated mental health was the primary outcome variable for the study. This variable has previously shown a positive correlation to other mental health morbidity measures (38), but should not be conflated with other more specific diagnostic categories such as depression or anxiety (39,40). Indeed, as a more global and subjective measure, many authors consider self-rated mental health as a more holistic measure of mental health outcomes which allows for a broad range of mental health issues to be captured (38,41), including mental health problems that are developing but which are not captured by more clinical mental health indicators (40).”
(line 146) report the question asked to the participants to evaluate the general sense of the state of mental health
Response: We have provided the exact wording and response options for the mental health variable, as provided below”
“Participants evaluated their current mental health on a Likert scale (At the present time, would you say your MENTAL HEALTH is: “Poor”, “Fair”, “Good”, “Very good”, or “Excellent”) (see Supplemental File 1). The variable was dichotomized to “Negative Self-Rated Mental Health” (“Poor” and “Fair”) and “Positive Self-Rated Mental Health” (“Good”, “Very good”, and “Excellent”). This was deemed to be an acceptable (if not conservative) approach to capture a general sense of mental health status based on precedent from previous studies using self-rated mental health (42) – allowing us to explicitly identify factors associated with sub-optimal (i.e., fair or poor) mental health.
(line 241-242) are you sure that there is a significant association with such a low percentage of "non-binary" participants (2.6%) "Other significant bivariable associations with negative self-assessed mental health were [] being not binary or a woman (against a man). "): check the analyzes again
Response: A sample size of 41 participants for the non-binary category and an overall sample size of 1,576 provides sufficiently narrow confidence intervals and small enough p-values to conclude statistical significance, despite the small overall proportion of respondents in this category. This is because the tests used rely on frequencies, not proportions, in the construction of confidence intervals. We have reported confidence intervals in table 2 to ensure readers are aware of the range of magnitudes of effect that can be calculated. As you will notice, the confidence intervals are considerably larger for the smaller groups, but they remain statistically significant due to the magnitude of the effect.
(242-243) are you sure that there is a significant association with such a low percentage of "African, Caribbean, or Black ethnicity" participants (2%) (the text states "self-rated mental health was significantly associated with African, Caribbean or black (vs. white) ethnicity "): check the analyzes again
Response: As with comment C.3., a sample size of n = 158 for African, Caribbean, and Black respondents does appear to be statistically significant due to the large effect size observed. The confidence interval captures the range of uncertainty, and is wider than the larger groups.
Reviewer 2 Report
Comments and Suggestions for Authors
The originality of this article resides in statistically analizing the differences in mental health between workers in public-facing occupations and those working from home, combined with the possible mediation effect of compliance with protective factors (COVID-19 vaccination, physical distancing, and mask adherence) on self-rated mental health of workers. That’s the main question the article addressed.
As the title indicate, it is a cross-sectional study that used secondary data collected from a survey. It is not an experimental research, but a naturalistic one, and the authors have capitalized the existent data to bring new and valuable information about the work place with regard to workers mental health, and in what extent the protective health measures influence such perception.
It adds to the area in terms of the timing (during the third wave in 2021), the occupational and ethnic diversity of the participants, and the inclusion of measures of the time of work from home, that are not frequent in previous studies. Also, for the testing of mediation from protective measures.
The authors set the question (104-105) and propose three hypothesis: Mental health is poorer in workers that no work from home (108); mental health will be better in those that work from home or in hybrid settings (113-114); and the no compliance with protective measures against COVID will mediated the relation between worse mental health and working from home (17-119).
It will improve the quality of the research to look for statistical differences by gender, given that previous research show that most of the domestic work and care rely on women. That modifies the stress they face at home and, consequently (we can hypothesize) their perception of mental health. The methodology is correct and the mediations analysis follows the step usually required and recommended.
The descriptive results will be better presented aligning the columns to the left in tables 1 and 2. Table 3 needs to be corrected, because the accidental inclusion of sequential numbers (617-671) create confusion.
The conclusions are coherent with the results and arguments presented, addressing the main question as well the hypothesis elaborated. Furthermore, the authors discuss the limitations of the study and propose new questions.
All the references included are relevant and pertinent to the subject.
I hope my remarks help the authors to improve their valuable work.

Author Response
It will improve the quality of the research to look for statistical differences by gender, given that previous research show that most of the domestic work and care rely on women. That modifies the stress they face at home and, consequently (we can hypothesize) their perception of mental health. The methodology is correct and the mediations analysis follows the step usually required and recommended.
Response: We agree with the reviewers and note that in the present study we controlled for the effect of gender, but effect modification was not explored. This is because we have planned future analyses with a more detailed set of mediators not included in this dataset focused on unpaid work. We have added note of this work in the “Future Research Directions” section of the manuscript, as follows:
“In the present study we did not account for important and salient factors such as living conditions, household composition, sources of material, social, and emotional support, non-work related labour, and other undoubtedly important factors. Future research will explore these factors in relation to working arrangements. Such analyses are critical for understanding the gendered dynamics of work from home. We hypothesize that this would be a critical moderator for exploration in future research. As well, family composition and income are critical moderators for understanding how people can best be sup-ported in distance work environments.”
The descriptive results will be better presented aligning the columns to the left in tables 1 and 2. Table 3 needs to be corrected, because the accidental inclusion of sequential numbers (617-671) create confusion.
Response: We have removed the erroneously included line numbers and fixed the alignment issues.
Reviewer 3 Report
Thank you for the opportunity to read this interesting manuscript. The topic is timely and needs to be researched before the impacts of COVID-19 are accepted as part of the 'new normal'. The manuscript could be further improved by a clearer explanation of how data gathered during the third wave relates to the sixth wave and the future. There were many contextual differences about the third wave which the authors discuss. The argument that makes this now historical data relevant in an ongoing sense needs to be made clearer. It is very difficult to separate out confounding variables in such complex situations. The statistical analysis does some of this and the authors also recognise the limitations of the research that was done. Did you examine other factors such as how home-schooling impacted on workers who had families/children? As an international reader, I would like more information about income brackets and local socio-economic factors in order to be able to interpret the data you have reported. I have made some specific comments below.
Page 2 line 70 – give full titles of GAD-2 and PHQ-2 screeners
Lines 75-76: Reconsider the wording – what do you mean by access to free…social distancing protocols”? Are protocols mandated by government or workplaces? Lines 78-79 then appear contradictory “we have also experienced a slow relaxation of public health orders…”
Lines 81-82: reword “pertinent need” as this is not semantically correct.
Lines 83 onwards: The tense is confusing. When was the ‘third wave’ in Canada? It may be clearer to use past tense consistently. This will help with the transition to discussion of sixth wave on line 92.
Lines 90 -91: Please provide citations to support statements about “mass deaths” etc.
Page 3, line 103: Authors state they are using data collected during the third wave. Please be consistent with past tense. Your intention seems to be to analysing data from third wave to find potential implications for the present and future waves. Tense is therefore important otherwise the research objective is unclear.
Page 4 line 169: Discussion of income ranges – is this Canadian or US dollars? It helps an international reader if currency is clarified and possibly discussed in US dollars. Please also indicate how and why these income brackets were chosen (i.e. low income according to Canadian government indicators?)
Page 6 table 1 – numbering is very confusing.
Author Response
The manuscript could be further improved by a clearer explanation of how data gathered during the third wave relates to the sixth wave and the future. There were many contextual differences about the third wave which the authors discuss. The argument that makes this now historical data relevant in an ongoing sense needs to be made clearer.
Response: Thank you for providing the opportunity to speak to this point. As this data is only 12-13 months old, we believe it remains highly relevant. Especially as risks for COVID-19 continue and many workplaces struggle with determining appropriate telework arrangements. Characterizing these issues, we have added the following discussion to our introduction section:
“Presently, at the end of the sixth wave of the COVID-19 pandemic has seen jurisdictions move further away from public health orders, following roll-outs of third doses for the majority of working age adults in response to the Omicron variant (34,35). It remains unclear how the ongoing need for vaccine uptake and the turbulent nature of the pandemic will impact mental health. Moreover, as many companies and organizations transitioned large numbers of staff to working from home or a hybrid of working from home and in-person work during earlier waves of the pandemic, this work will be relevant for both employers and policy makers respectively to assess the costs and benefits of different arrangements as workplaces largely return to in-person work. Determining the extent of any differences in mental health related to work-from-home status has clear health equity implications for employers and policy makers to ensure best practices throughout the on-going COVID-19 pandemic as well as for future public health crises. As COVID-19 risks continue to the present day – particularly with risks such as long-COVID and unmitigated Omicron infection – it has become important to understand mental health differences ac-cording to where participants are working."
Did you examine other factors such as how home-schooling impacted on workers who had families/children?
Response: We were not able to explore this question of how home-schooling impacted workers who had families/children as this was not a question asked in this year’s version of the CSCS. However, we agree that this is a limitation of our analysis and worthy of further study and have made note of this in our future research section:
“In the present study we did not account for important and salient factors such as living conditions, household composition, sources of material, social, and emotional support, non-work related labour, and other undoubtedly important factors. Future re-search will explore these factors in relation to working arrangements. Such analyses are critical for understanding the gendered dynamics of work from home. We hypothesize that this would be a critical moderator for exploration in future research. As well, family composition and income are critical moderators for understanding how people can best be supported in distance work environments.”
Page 2 line 70 – give full titles of GAD-2 and PHQ-2 screeners
Response: We have added in full titles for GAD-2 and PHQ-2 screeners.
Lines 75-76: Reconsider the wording – what do you mean by access to free…social distancing protocols”? Are protocols mandated by government or workplaces? Lines 78-79 then appear contradictory “we have also experienced a slow relaxation of public health orders…”
Response: Thank you for highlighting the lack of clarity here. We have revised this sentence as follows:
“Unlike during the first waves of the pandemic, Canadians now have access to free vaccines and masks; and other risk mitigation approaches (e.g., physical distancing, ventilation) are better understood by the public. These measures may, therefore, mitigate the fear of COVID-19 and its associated stress for people working in public, front-facing jobs (3).”
Lines 81-82: reword “pertinent need” as this is not semantically correct.
Response: We have removed the word "pertinent".
Lines 83 onwards: The tense is confusing. When was the ‘third wave’ in Canada? It may be clearer to use past tense consistently. This will help with the transition to discussion of sixth wave on line 92.
Response: Thank you for this comment. We have revised the tense so that it is referred to in the past tense.
Lines 90 -91: Please provide citations to support statements about “mass deaths” etc.
Response: We have removed the term “mass deaths” and replaced it with “deaths.” As Well, we have added the following citations to support statements about the overwhelmed healthcare system, deaths in LTC, and socially isolating lockdown measures:
Best LA, Law MA, Roach S, Wilbiks JMP. The psychological impact of COVID-19 in Canada: Effects of social isolation during the initial response. Canadian Psychology / Psychologie canadienne. 2021;62(1):143.
Canadian Institute for Health Information. COVID-19’s impact on hospital services [Internet]. 2021 [cited 2022 Aug 23]. Available from: https://www.cihi.ca/en/covid-19-resources/impact-of-covid-19-on-canadas-health-care-systems/hospital-services
Canadian Institute for Health Information. COVID-19’s impact on long-term care [Internet]. 2021 [cited 2022 Aug 23]. Available from:
Page 3, line 103: Authors state they are using data collected during the third wave. Please be consistent with past tense. Your intention seems to be to analysing data from third wave to find potential implications for the present and future waves. Tense is therefore important otherwise the research objective is unclear.
Response: We have revised the tense for consistency.
Page 4 line 169: Discussion of income ranges – is this Canadian or US dollars? It helps an international reader if currency is clarified and possibly discussed in US dollars. Please also indicate how and why these income brackets were chosen (i.e. low income according to Canadian government indicators?)
Response: The values are in CAD dollars. We have added the CAD indicator to values reported in texts and tables. To measure income, participant’s household income was collected in increments of $10,000 CAD. Bins of $30,000 CAD were selected with consideration of classifying individuals according to approximate thresholds for low- (e.g., Approx. $30,000 per households) and median income (approx. $90,000 per household) in Canada. As household size and cost-of living values varied, a more nuanced measure of income would have been preferred by was not available. We have added these issues to the limitations section and thank the reviewer for highlighting these questions.
“Likewise, some measures need refined assessment in future studies. For example, to measure income, participant’s household income was collected in increments of $10,000 CAD. Bins of $30,000 CAD were selected with consideration of classifying individuals according to approximate thresholds for low- (e.g., Approx. $30,000 per households) and median income (approx. $90,000 per household) in Canada. As household size and cost-of living values varied, a more nuanced measure of income would have been preferred by was not available in this secondary data analysis. Personal income, adjusted for cost of living, could provide a more nuanced insight into working condition and types of work engaged in – which are undoubtedly important for understanding worker health.”
Page 6 table 1 – numbering is very confusing.
Response: We are confused as well by the addition of these numbers in the version of the draft received by reviewers. We have removed the row numbers from all tables.
Reviewer 4 Report
Comments to Authors
Recommendation: Reject
First, I would like to thank the Editor for trusting me with the opportunity to review this research, "The Impact of Working from Home on Mental Health: A Cross-Sectional Study of Canadian Worker’s Mental Health During the Third Wave of the COVID-19 Pandemic." The theme of the study seems interesting and has the potential to make a contribution; however, except for the introduction section, the rest of the manuscript is written haphazardly. The authors have articulated the introduction very well while building upon the problem and the need of the study. There are several other reasons that compelled to warrant a rejection. The detailed comments are as below:
1- The introduction is well written and arguably builds upon the problem and the need of the study.
2- As the study is empirical in nature and tests the hypotheses applying logistic regression models; but nowhere the study mentions any proposed hypotheses. The entire review and hypotheses development section is missing from the manuscript. The authors must come up with a “Theory and Hypotheses Development” section before jumping to methods sections and arguably build the literature in support of the proposed hypotheses while being critical of the previous literature. Some of the recent studies from the domain of telecommuting research are listed as follows;
Jamal, M.T., Anwar, I., Khan, N.A. and Saleem, I. (2021), "Work during COVID-19: assessing the influence of job demands and resources on practical and psychological outcomes for employees", Asia-Pacific Journal of Business Administration, https://doi.org/10.1108/APJBA-05-2020-0149
Hayes, S. W., Priestley, J. L., Moore, B. A., & Ray, H. E. (2021). Perceived Stress, Work-Related Burnout, and Working From Home Before and During COVID-19: An Examination of Workers in the United States. SAGE Open, 11(4), 21582440211058190.
Jamal, M. T., Alalyani, W. R., Thoudam, P., Anwar, I., & Bino, E. (2021). Telecommuting during COVID 19: A Moderated-Mediation Approach Linking Job Resources to Job Satisfaction. Sustainability, 13(20), 11449. https://doi.org/10.3390/su132011449
Jamal, M. T., Anwar, I., & Khan, N. A. (2021). Voluntary part-time and mandatory full-time telecommuting: a comparative longitudinal analysis of the impact of managerial, work and individual characteristics on job performance. International Journal of Manpower. https://doi.org/10.1108/IJM-05-2021-0281
Kumar, P., Kumar, N., Aggarwal, P., & Yeap, J. A. L. (2021). Working in lockdown: The relationship between COVID-19 induced work stressors, job performance, distress, and life satisfaction. Current Psychology, 40(12), 6308–6323.
3- The use of Logistic regression seems fine because the outcome variable was dichotomized into two categories “Negative Self-Rated Mental Health” and “Positive Self-Rated Mental Health.” However, I have my apprehensions about testing the mediation effect using Logistic Regression while drawing support from Barron and Kenny (1986) approach, which purely relies on the linearity concept, whereas logistic regression is a non-linear model. I suggest the authors to consider the following sources:
4- The reporting of the results is haphazard and does not meet the reporting standards of JARS (https://apastyle.apa.org/jars). The tables are illegible because of the insertion of unnecessary line numbers.
5- The discussion section is somewhat too general and lacks specificity.
6- The study does not offer any implications for the theory and practice. The authors are suggested to provide a separate section discussing the implications of the findings for both theory and practice.
13- The manuscript needs grammar check and proofreading.

Author Response
2- As the study is empirical in nature and tests the hypotheses applying logistic regression models; but nowhere the study mentions any proposed hypotheses. The entire review and hypotheses development section is missing from the manuscript. The authors must come up with a “Theory and Hypotheses Development” section before jumping to methods sections and arguably build the literature in support of the proposed hypotheses while being critical of the previous literature.
Response: We thank the reviewer for their note. We’ve ensured the presence of a hypothesis section at the end of our introduction section. This section includes our hypotheses for the regression and mediation analyses. This section reads as follows:
“This study used survey data collected during the third wave of the COVID-19 pandemic in Canada (36) to examine whether there were any differences in self-rated mental health based on work setting and if so, what contributes to these differences? The dataset provided a unique opportunity to explore the nuances of self-rated mental health, and thus, bivariable and multivariable logistic regression models were used to test the hypothesis that mental health status is poorer among individuals who are not working from home. Additionally, physical distancing and mask wearing, which have been common practice since the pandemic’s onset, will be tested as mediators due to their potential for combating pandemic-related stressors related to concerns about COVID-19 transmission (37). A mediation analysis tested whether COVID-19 vaccination, physical distancing, and mask adherence – due to their effectiveness as COVID-19 mitigation measures – had significant and protective effects on self-rated mental health. In conducting these analyses, we hypothesized that people working from home or engaging in hybrid work arrangements had better self-rated mental health than those working exclusively in-person. We further hypothesized that the exposure to COVID-19, as reflected in lack of compliance with public safety COVID-19 prevention guidelines, would partially mediate the association between working from home and worse self-rated mental health.”
3- The use of Logistic regression seems fine because the outcome variable was dichotomized into two categories “Negative Self-Rated Mental Health” and “Positive Self-Rated Mental Health.” However, I have my apprehensions about testing the mediation effect using Logistic Regression while drawing support from Barron and Kenny (1986) approach, which purely relies on the linearity concept, whereas logistic regression is a non-linear model. I suggest the authors to consider the following sources: http://davidakenny.net/doc/dichmed.pdf https://web.pdx.edu/~newsomj/cdaclass/ho_mediation.pdf
Response: We have clarified that the mediate() package was used for the actual analyses, but that the Barron and Kenny process was used to examine the relationships between explanatory, outcome, and mediating measures:
“Mediation analysis was followed firstly via Baron and Kenney’s (1986) steps for deter-mining mediation via logistic regression models and secondly by utilizing the mediate package in R with bootstrapping enabled (49,50). The mediate package explicitly allows for handling of binary and logistic measures outside of a linear framework, while Baron and Kenney’s (1986) steps provide a process for reviewing bivariable and multivariable models, which has helped us to evaluate the associations between our primary exposure and outcome, primary exposure and mediator, mediator and outcome, and primary ex-posure while controlling for the mediator and outcome. The mediate function was then used for more rigorous tests of indirect (mediation) effects on the outcome variable (50).”
4- The reporting of the results is haphazard and does not meet the reporting standards of JARS (https://apastyle.apa.org/jars). The tables are illegible because of the insertion of unnecessary line numbers.
Response: The line numbers have been removed and we have reviewed our tables to ensure they provide all necessary information (per ACS format, as required by this journal). If the editor feels that additional information is needed, we are happy to update the reported results. However, for our discipline, the reported information seems standard.
The discussion section is somewhat too general and lacks specificity. The study does not offer any implications for the theory and practice. The authors are suggested to provide a separate section discussing the implications of the findings for both theory and practice.
Response: Throughout the discussion section, we discuss potential implications of our findings. However, with consideration to the state of the literature we do not feel we can offer a conclusive implications for practice. We do agree that implications for theory and research would be helpful and have added a future research directions section:
“Recognizing these limitations, as well as several opportunities to establish new lines of inquiry, we recommend that future research on the COVID-19 pandemic and future communicable disease epidemics should aim to sample a more representative group of people working from home; determine interactions between ethnic, sexual and gender minorities, and older populations; and incorporate measures of self-assessed psychological distress around workplace safety. Furthermore, as noted above, the present study did not account for important and salient factors such as living conditions, household composition, sources of material, social, and emotional support, non-work related labor, and other undoubtedly important factors. Future research will explore these factors in relation to working arrangements. Such analyses are critical for understanding the gendered dynamics of work from home. We hypothesize that this would be a critical moderator for exploration in future research. As well, family composition and income are critical moderators for understanding how people can best be supported in distance work environments. Future research should therefore, conduct more narrow analyses or improve measurements of these key factors so that a more nuanced profile of working conditions (e.g., income, class, status, hierarchy) can be assessed in relation to our research questions. Finally, it is critical for longitudinal within person studies to continue examining the effect of work from home on individual health and wellbeing. “
13- The manuscript needs grammar check and proofreading.
Response: Thank you for raising this concern. We have extensively proofread and checked the manuscript for any errors.
Round 2
Reviewer 4 Report
Comments to Authors
Recommendation: Reject
With greetings of the day!
Emphasizing the serious flaws the study suffered with. Moreover, despite providing detailed comments highlighting the theoretical and methodological flaws, the authors have not addressed the comments in the revised version, and the manuscript remains almost the same. Neither review section has been added, nor hypotheses have been clearly mentioned. My apprehensions still remain the same regarding testing mediation analysis with a binary outcome variable. The authors have claimed to use ‘Mediation’ package in R for mediation analysis; however, this R package is also for causal mediation, while study’s outcome variable is binary and hence does not lead to causality. An appropriate method for testing the mediation with a binary outcome variable would be the Generalized Linear Mediation Model (Preacher, 2015, page 839).
Preacher, K. J. (2015). Advances in mediation analysis: A survey and synthesis of new developments. Annual review of psychology, 66, 825-852.
Further, the authors were suggested to revise the discussion section making it more problem-specific while corroborating with the previous literature. However, nothing has been done by the authors, and the discussion section is still the same.
As the study (previous version) did not offer any implications for the theory and practice, I suggested incorporating a separate section signifying the implications of the study’s findings for the theory and practice. But the authors again overlooked the suggestion and did not add an implications section in the revised manuscript.

Author Response
1) We disagree with the reviewer regarding their conclusion that the mediate() function does not handle binary mediators. As per the documentation, “The mediate function automatically detects the type of models used for the mediator and outcome models and calculates the estimates of the ACME and other quantities of interest via the general algorithms described in Imai et al. (2010a).” (See https://cran.r-project.org/web/packages/mediation/vignettes/mediation.pdf)
Second, the reviewer claims our introduction and discussion are insufficient, but does not provide enough detail or any changes. They claim that we have not provided hypotheses, which we quoted directly in our last response. We also cited several sections that discuss the implications of our work. These lines of critique are vague and leave little to be engaged with. It is our opinion that we have provided a robust introduction and discussion sections. This is evident from the comment of other reviewers who described our paper as “well-written”, “relevant”, and “timely.” It is clear from these reviews that the other reviewers have engaged with our work in a meaningful way and do not share the opinion of reviewer 4. In particular, reviewer 2 confirms that we have provided discussions of limitations, appropriate references, a discussion of our hypotheses and main questions. As such, we do not feel the changes are necessary since the suggestions are merely stylistic preferences of the reviewers and not specific enough critiques as to merit revisions.